

# Bacterial community diversity, lignocellulose components, and histological changes in composting using agricultural straws for *Agaricus bisporus* production

Tingting Song[1], Yingyue Shen[1], Qunli Jin[1], Weilin Feng[1], Lijun Fan[1], Guangtian Cao[2] and Weiming Cai[1]

[1] Institute of Horticulture, Zhejiang Academy of Agricultural Sciences, Hanzghou, Zhejiang, China
[2] China Jiliang University, College of Standardisation, Hangzhou, Zhejiang, China

## ABSTRACT

Agricultural straws (AS) may serve as potential base-substances in the production of *Agaricus bisporus*. Six AS that occur across China were investigated in a two-stage composting experiment; lignocellulose components, AS morphology, and the effects of different AS on mushroom yields from 2015–2017 were examined. In addition, microbial biodiversity and their impact on substrate degradation were studied using 16S gene sequenc based on six different AS on the 3rd (I.F), 6th (I.S), and 10th (I.T) day of Phase I, and Phase II (II). Results showed that the six different AS exhibited differences in the progression of degradation under the same compost condition; the wheat straw, rice straw, and cotton straw induced a significantly higher mushroom yield than did the others ($P < 0.05$); *Thermobispora*, *Thermopolyspora,* and *Vulgatibacter* genera may play an important role in the different AS degradations. According to our experiments, we can adjust formulations and compost methods to obtain high-yield mushroom compost based on different AS in the future.

## INTRODUCTION

Agricultural straws (AS) are organic agricultural residues produced at a global rate of $2–4 \times 10^9$ t per year, which is 1.4 times the annual crop yield (*Smil, 1999*). AS comprise mainly of lignocellulosic biomass, most of which is used for combustion in China (*Sheng et al., 2018*). Until 2018, the amount of AS had reached 886 million tons in China. AS waste is rich in macromolecular organic matter, including cellulose, hemicellulose, starch, and lignin. Many techniques have been applied to the utilization of AS waste resources. Among these methods, mushroom cultivation is one of the most efficient and economical biotechnological process for converting lignocellulosic materials into high quality food. Rice straw and wheat straw have been used for hydrogen production (*Liu et al., 2015*) and ethanol fermentation (*Nair et al., 2017*).

Corresponding authors
Guangtian Cao,
15a1903025@cjlu.edu.cn
Weiming Cai, caiwm0527@126.com

White button mushrooms (*Agaricus bisporus*) are well-known basidiomycete fungi, grown on substrate with a casing layer and produced in large quantities for human consumption (*De Groot et al., 1998*). In America and Europe, the wheat straw is a major traditional ingredients for *A. bisporus* cultivating (*Straatsma et al., 2000*). In China, rice straw is widely available, and has formed the major material for *A. bisporus* cultivation in recent decades (*Gong, 2011*). There are three distinct phases of the compost production and preparation (*McGee, 2018*): Phases I and II form the aerobic fermentation processes (*Simsek et al., 2008*), and Phase III form inoculation and mycelium growth processes (*Patyshakuliyeva et al., 2015*). In general, Phase I periods of about 5–7 and 15–21 days are used in Europe and China, respectively. The compost temperature stays at approximately 80 °C due to the growth of thermophilic microorganisms (*Zhang et al., 2019*). Subsequently, a pasteurization process (Phase II) is performed at 45–50 °C for about 4–9 days until the ammonia level decreases to a level that is non-toxic to *A. bisporus* mycelia. The compost preparation is achieved by manipulating the natural succession of microorganisms present in the raw materials. The microbial populations, substrate compositions, and the physical states of the materials change during Phases I and II of this process.

The bacterial community of the compost is highly diverse, and exhibits dynamic changes in response to the quality of the compost (*Székely et al., 2009*; *Zhang et al., 2014*; *Siyoum et al., 2016*; *McGee et al., 2017*). The substrates are self-heated after mixing and wetting in Phase I compost and are characterized by high microbial activity (*Miller, Harper & Macauley, 1989*), while aerobic microorganisms dominate the fermentation process because of controlled temperature conditions in Phase II compost (*Straatsma et al., 1995*). For the past decades, several techniques have been used for studying of microbial communities at different stages, including community-level physiological profile analysis, phospholipid fatty acid analysis, and environmental DNA amplification-based methods (*Dees & Ghiorse, 2001*; *Tiquia & Michel, 2002*; *Helgason, Walley & Germida, 2010*; *Takaku et al., 2006*). The development of next-generation sequencing technologies has provide a more reliable method for the analysis of the depth and composition of microbial communities.

In order to make full use of AS across the different areas of China, six major straws and residues, including wheat straw (WS), rice straw (RS), cotton straw (CS), corncob (CC), corn straw (C), and bagasse (B), were selected as the representative substrates to cultivate *A. bisporus,* following the initial screening of various materials. Then, physicochemical properties, histological changes, and the production properties of the different *A. bisporus* compost materials were compared. Additionally, the bacterial community structure of four-stage composting, based on six different AS, was analyzed. The data obtained by our experiment may help understand the optimal composting conditions corresponding to certain AS formulations. The efficient utilization of different AS from different areas as substrates for *A. bisporus* cultivation has the potential to reduce the cost of transporting raw materials and increase farming profits in China.

## MATERIALS AND METHODS

### Mushroom strain

*The A. bisporus* strain A106 was obtained from our lab and cultured at 25 °C on potato dextrose agar (PDA) medium for further experiment. Wheat grains (purchased from a local grocery store ) were used to prepare *A. bisporus* spawn as described by *Cheng (1999)*.

### Composting and sampling

Composting was conducted in the fermentation room of our experimental base located in Hangzhou, Zhejiang province, between 2015 and 2017. The composting windrows (5 × 2.0 × 1.8 m) each contain a 600 kg mixture consisting of Agriculture straw, rapeseed cake, $CaSO_4 \cdot 2H_2O$, $Ca(H_2PO_4)_2 \cdot H_2O$, $(NH_4)_2SO_4$, and urea ($CON_2H_4$), with the formulations of ingredients listed in S1.

Before composting stage, the raw material were mixed and wetted by manual spraying to adjust the moisture around 75%, and ∼200 g samples (M) were collected in triplicate. At Phase I of the 3 th (I.F), 6 th (I.S), and 10 th (I.T) day, the windrows were turned on to keep the concentration of $O_2$ sufficient and enhance the composting process. $CaCO_3$ and water were added manually to maintain the pH (6–8) and moisture content (60–70%) during the turning (data in S2). Phase II (II), consisting of 6 days was characterized by a rapid increase in temperature up to 60 °C for 8–9 h, followed by stabilization of the compost temperature to 45–50 °C for 5 days, and gradual cooling to 25 °C. Samples at four different stage (∼200 g each) were collected and dried using a vacuum freeze dryer as previously described (*Song et al., 2014*), then stored at −80 °C for further analysis.

### Analysis of physical and chemical properties

Throughout Phase I and II, the temperature of the tunnel was monitored by tubular mercurial thermometers continuously placed in the middle of the compost (Kai Longda Instrument Co., Tianjin, China).The water content was measured by water analyzer (Ohaus, MB25) with 5 g sub-samples. The pH values were determined with an electronic pH meter (Mettler-Toledo Instruments Co., Ltd., Shanghai, China) using 10% (w/v) sample suspensions (in distilled water) were used. The total nitrogen (N) and carbon (C) levels were assayed by the methods of Kjeldahl (*Bremner & Mulvaney, 1982*) and $K_2Cr_2O_3$ oxidation (*Fan, 2007*), and then C/N ratios were calculated. In addition, the fiber profiles (including crude fiber of the cell wall, cellulose, hemicellulose, and acid digestible lignin [ADL]) were analyzed as described by *Van Soest, Robertson & Lewis (1991)*. The percentage of dry matter losses was calculated according to the following equations: Dry matter loss (%) = (the dry matter weight after Phase II/the dry matter of initial formulation) × 100%.

### AS stem sub-samples for transmission electron microscopy

The sub-stem samples (three cm long) were obtained from dried samples of six different AS composts collected after Phase II. Each sample was prepared and subjected to the method described by *Song et al. (2014)*. The samples were examined using a JEM-1230 transmission electron microscope (TEM).

## Determination of mushroom yields

The mushroom yields were recorded from 2015–2017 for three years. The pawning rate of six different pasteurized compost was 1% (w/w) . Then, the spawned compost was divided into 15.0 kg samples (wet weight, moisture content ∼65%) and transferred to a 45–35 × 24 cm plastic cultivation boxes. After growing the mycelia at 22–25 °C for 25 days, the casing soil was overlaidto initiate primordium formation. Fruit bodies in different AS composts were harvested, and the weights of the total fresh mushroom yields (TY, g/kg substrate) during the harvesting period, as well as the 1–3 flushes (T3) of fruit bodies were recorded. Biological efficiency (BE) values were calculated by the ratio of fresh mushroom weight to dry substrate weight. All experiments in each year included 10 replicates for each treatment, and the yields of mushroom cultured with different AS-based substances were calculated by the average of the three-year data from 2015–2017.

## High-throughput sequencing of 16S rRNA

Total genomic DNA was extracted from 1.0 g mixture of AS (3 collection points in 2017), using the Power Soil DNA Extraction Kit (Mo Bio Laboratories Inc., Carlsbad, CA, USA) as per the manufacturer's instructions. The 16S rRNA amplicon sequencing was performed according to our previous study (Cao et al., 2018); briefly, sequencing was performed on a Illumina MiSeq platform at Novogene Cooperation (Beijing, China) with a HiSeq 2500 (PE250) sequencing system, using the specific primer pair of 515F (5′-GTGCCAGCMGCCGCGGTAA-3′) and 806R (5′-GGACTACHV GGGTWTCTAAT-3′), targeting the V3–V4 hypervariable region. The data that supporting this study are openly available in the NCBI GenBank (BioProject accession number: PRJNA664798).

## Statistical analysis

The data are recorded as the means of replicates with standard error(±SE). SPSS 16.0 software was used to analyze the data through one-way and two-way ANOVA, and $P < 0.05$ were considered to indicate significant differences using the Tukey test. For bioinformatics analysis, as per our previous study (Cao et al., 2018), the similarity or difference between the different sequencing libraries was assessed by the α- (Shannon and Simpson indices) and β-diversity parameters, with the former analyzed by the Wilcoxon signed-rank test and the latter by the weighted UniFrac analysis (QIIME). Principal coordinate analysis (PCoA) and non-metric multidimensional scaling (NMDS) plot analysis were performed using the R software (Version 2.15.3). Linear discriminant analysis with effect size (LEfSe) tool was accessed online at http://huttenhower.sph.harvard.edu/galaxy/. The student $t$-test was used to analyze the mean difference between two samples of bacterial strains at the genus level for the three stages of Phase I and II, respectively.

# RESULTS

## Temperature change in the fermentation chamber from 2015–2017

Figure 1 shows that the mean compost temperatures in the fermentation chamber fluctuated daily between 40.1 and 72.8 °C during the composting period from 2015 to 2017. As expected, the compost temperatures decreased rapidly on the 4th, 6th and 9th days due
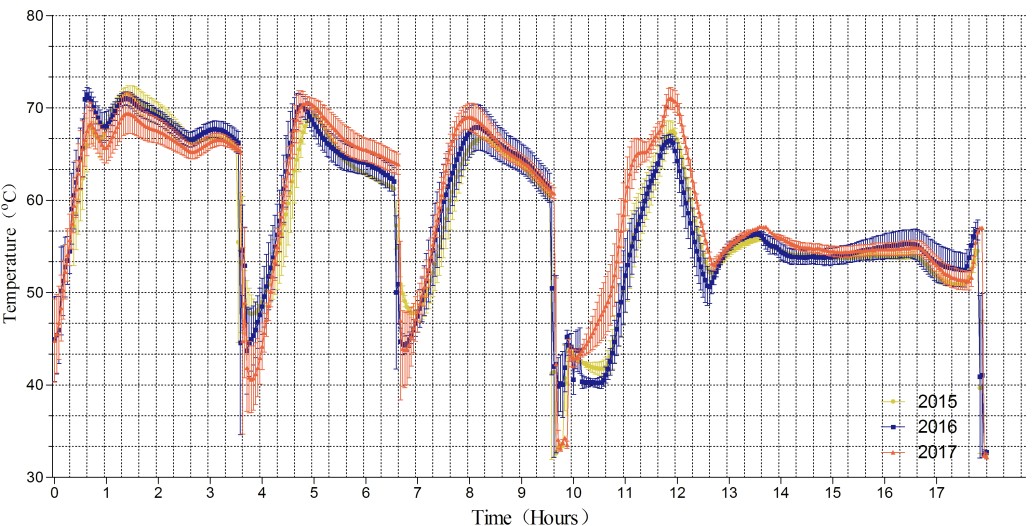

**Figure 1** Temperature change of fermentation chamber.

to turning. We found that the temperature was maintained at approximately 65–70 °C over the first 3 days, decreased from 70 °C to 65 °C on days 5–6 and 8–9 respectively, and then finally to approximately 55° C from day 13 to the end of the trial. Because this trial was performed in temperature-controlled tunnel, there was no dramatic difference in temperature in the fermentation chamber between from 2015–2017; this enabled stable composting over the three-year period.

## Physical characteristics of AS-based substances

After the two phases of composting, the C/N ratios of the initial formulations of six different AS-based substances, which were 29.6–32.8:1, decreased to final 15.1–19.2:1. There was no significant difference in the C/N ratios among the different AS-based substances across all stages (Fig. 2A). The data showed no remarkable difference in the crude fiber content among all samples at all time points apart from CS , which had a significantly higher crude fiber content than WS in Phase II (Fig. 2B). The data showed that the rice straw (RS)-based substrate had a much greater nitrogen content than did the wheat straw (WS) and corncob (CC) based substrates in the initial formulation stage. Moreover, RS had a significantly higher nitrogen composition than did CC in the Phase I.F , while both cotton straw (CS) and CC had a significantly higher nitrogen composition than bagasse (B) in Phase II (Fig. 2C).

Furthermore, the percentages of cellulose, hemicellulose, and ADL in all phases were examined (Fig. 3). After Phase II, all substances sampled had significantly less hemicellulose ($P < 0.05$) than that in the material phase, and samples of CS, WS, RS, and CC all had significantly less cellulose ($P < 0.05$); no significant differences were found in the ADL percentages across all samples (Figs. 3B–3D). The dry matter loss was also analyzed at the end of the trial, and we found that the CC had the lowest percentage loss, while CS and B had the highest.

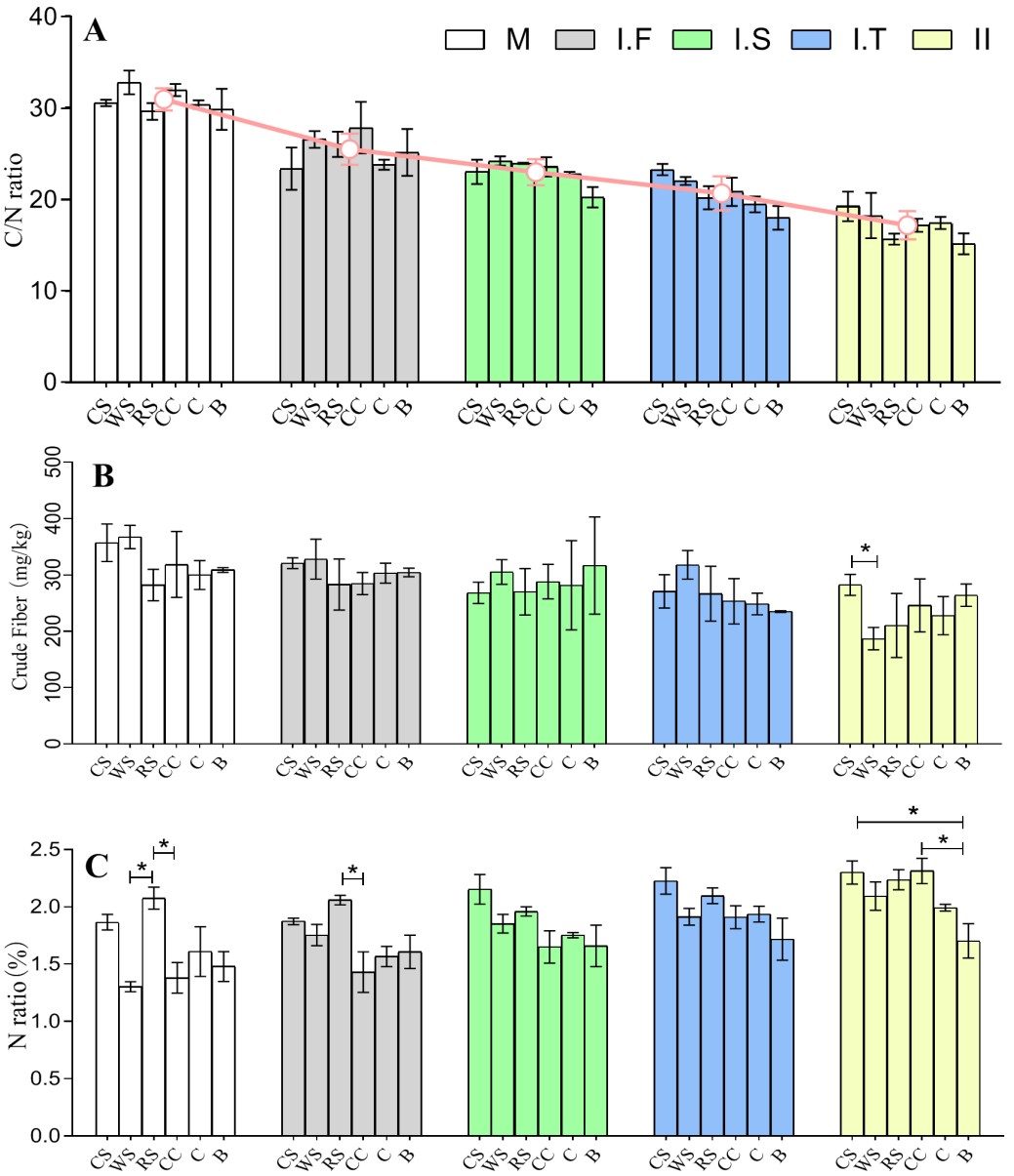

**Figure 2** **Chemical properties of six different AS compost changes in the C/N ratio (A), crude fiber content (B), and the total N amount (C) from the M to Phase II.** The red line in (A) represents the average changes of six different AS in C/N ratio at different stages of fermentation. WS: wheat straw substrate; RS: rice straw substrate; CS: cotton straw substrate; CC: corncob substrate; C: corn straw substrate; B: bagasse substrate. The C/N ratio was obtained based on total nitrogen according to the Kjeldahl method and carbon content according to the $K_2Cr_2O_3$ oxidation method. The total N is expressed as a ratio of protein in 100 g dry material.

## TEM scanning of AS morphology

The changes in AS structure from the material phase and Phase II were revealed by TEM scanning, as shown in Fig. 4. Compared to the straw materials of CS, CC, C, and B (Figs. 4A, 4G, 4H, and 4I), the cellular structure of straw in Phase II (Figs. 4D, 4J, 4K, and 4L) was

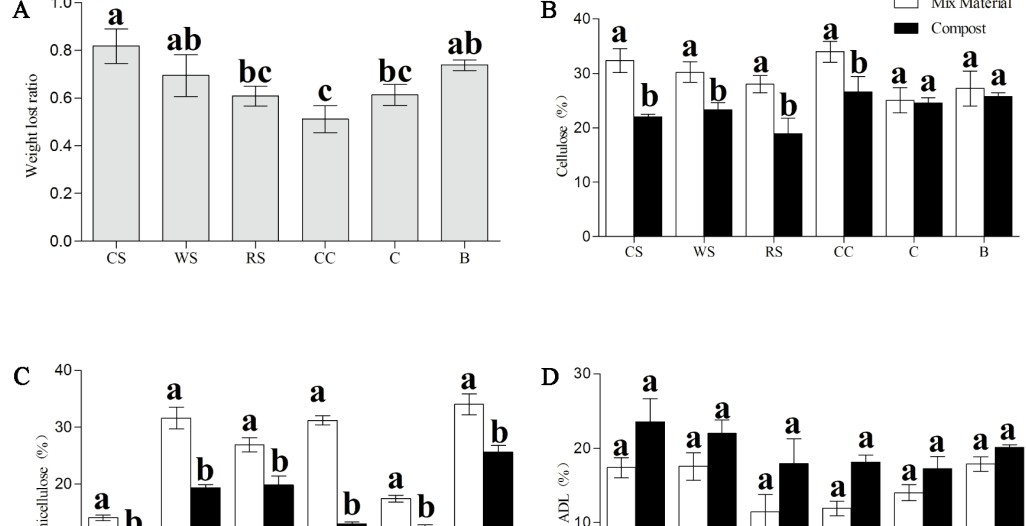

**Figure 3** Change in the dry matter loss (A), and contents of cellulose (B), hemicellulose (C), and ADL (D) of AS substances between material phase and Phase II.

dramatically destroyed, with numerous erosion troughs and cracks found on the surface of pretreated straws. This was particularly evident in the structures of C and CC, where the straw was severely destroyed and no intact cell wall structures were observed. The cellular structure of WS and RS appeared deformed to a certain extent (Figs. 4B & 4E, and 4C & 4F), which crimped cell wall generation.

## Yields of mushroom cultured with AS-based substances from 2015–2017

The mushroom yields cultured with AS-based substances from 2015–2017 are shown in Table 1 (the average of the three year period)and S3. According to the data analysis, There was no difference among the there-years mushroom yields in TY ($F_{2,17}$=0.15 $P = 0.86$), T3 ($F_{2,17} = 0.56$ $P = 0.59$), and BE ($F_{2,17} = 0.77$ $P = 0.49$), indicating that the three-year production data are relatively stable. From the TY, the CS (596.7 g/kg) substance was significantly higher than that in other groups, with RS (569.9 g/kg), WS (553.8 g/kg), and CC (498.0 g/kg) being higher than C (370.5 g/kg) and B (318.7 g/kg). The T3 performance was the highest in CS (501.2 g/kg), WS(478.6 g/kg) and RS (481.9 g/kg), and the lowest in C and B. We next analyzed the BE index for explaining the mushroom production characteristics of the different materials. The data showed that the highest BE value obtained for CS, WS and RS was 84.4,83.5 and 86.8 respectively, while C (63.7%) had the lowest value.

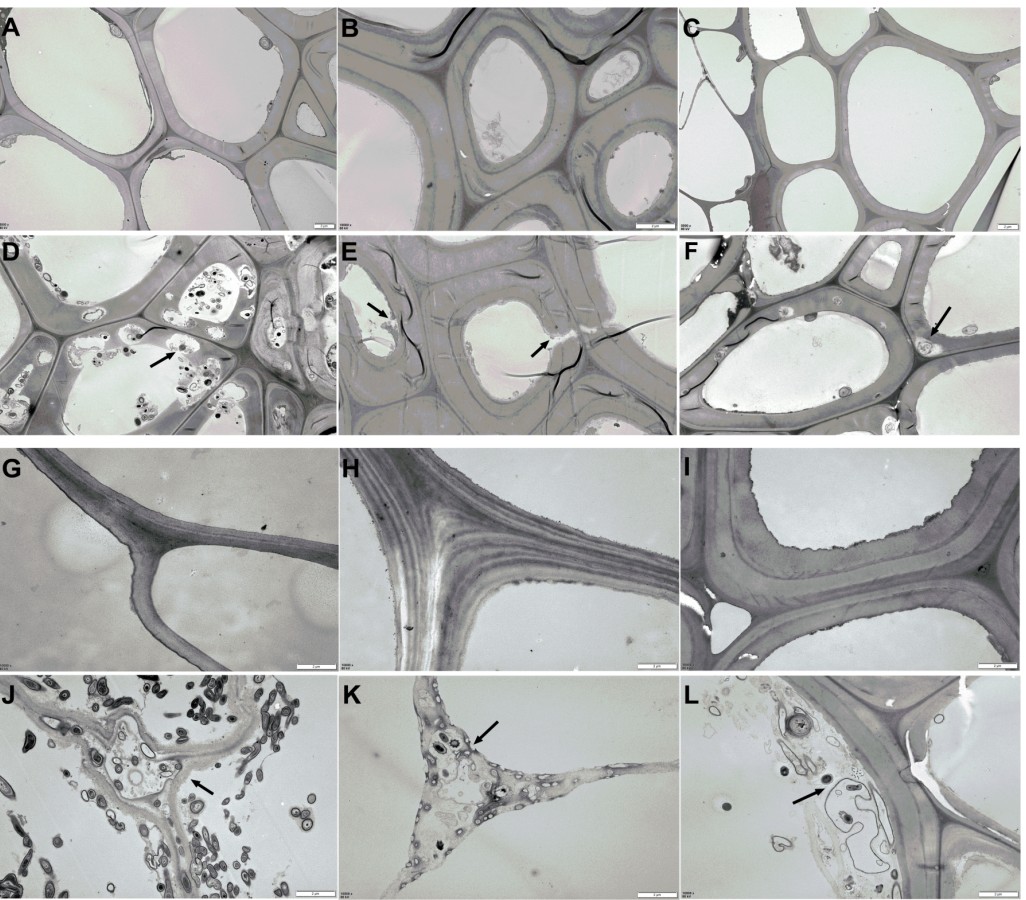

**Figure 4** **TEM images of AS structural changes after composting.** (A–C and G–H) The microstructure of the original AS material; (D–F and J–K) the AS microstructure after Phase II composting: (A and D) the CS raw and composting materials, respectively; (B and E) WS raw and composting materials, respectively; (C and F) the RS raw and composting materials, respectively; (G and J) the CC raw and composting materials, respectively; (H and K) the C raw and composting materials, respectively; (I and L) the B raw and composting materials, respectively.

## 16S High throughput sequencing results

The six AS under the same composting conditions were sampled as replicates; changes in the compost microflora community in different phases are shown in Figs. 5 and 6. Approximately 1,899 common operational taxonomic units(OTUs) were shared between all samples, while 1,485, 1,157, 1,170, and 1,024 specific OTUs were found in I.F, I.S, I.T, and Phase II, respectively (Fig. 5A). Firmicutes, Proteobacteria, and Actinobacteria were the three predominant phyla in all samples (Fig. 5B). In addition, Bacteroidetes dominated the samples of Phase I.F , and Gemmatimonadetes dominated the Phase II (Fig. 5C). The top 5 dominant genera of samples in the Phase I.F and I.S were *Pseudoxanthomonas*, *Thermobispora*, *Thermopolyspora*, *Thermobifida,* and *Thermobacillus*; *Thermobispora*, *Pseudomonas*, *Thermopolyspora*, *Thermobifida,* and *Ruminiclostridium* were the top 5 genera in the PhaseI.T; and *Thermobispora*, *Thermopolyspora*, *Thermobifida*, *Microbispora*

**Table 1** Analysis of mushroom yield (*Agaricus bisporus*) cultured on six different AS-based substances.

| Item | The average of three-year(Mean ± SE)[a] | | |
|---|---|---|---|
| | TY (g/kg) | T₃ (g/kg) | BE(%) |
| Cotton straw substrate(CS) | 596.7 ± 14.23[a] | 501.2 ± 11.30[a] | 84.4 ± 3.35[a] |
| Wheat straw substrate(WS) | 569.9 ± 27.15[ab] | 478.6 ± 50.58[a] | 83.5 ± 5.12[a] |
| Rice straw substrate(RS) | 553.8 ± 11.25[ab] | 481.9 ± 27.29[a] | 86.8 ± 3.33[a] |
| Corncob substrate(CC) | 498.0 ± 19.86[b] | 379.5 ± 19.63[ab] | 76.1 ± 1.80[ab] |
| Corn straw substrate(C) | 370.5 ± 12.56[c] | 233.8 ± 6.56[c] | 63.7 ± 2.104[b] |
| Bagasse substrate(B) | 318.7 ± 16.25[c] | 250.6 ± 17.50[bc] | 78.9 ± 2.0[ab] |
| $F_{(5,17)}$[b] | 35.75[**] | 19.36[**] | 6.80[**] |

Notes.

[a] Data were expressed as mean standard error ($n = 3$). The data in the same column marked with different lowercase letters are significantly different (Tukeys HSD, $p < 0.05$).

[b] $F$ value in one-way ANOVA:

[*] $p < 0.05$.

[**] $p < 0.01$

and *Thermobacillus* were the top 5 genera of Phase II. Moreover, the α-diversity showed that the Shannon and Simpson indices of I.S samples were significantly higher than those in Phase II samples; no remarkable differences were found among all other samples (Figs. 5D and 5E). Both PCoA and NMDS analysis showed that the samples of Phase II were distinctly separated from other samples of Phase I (Figs. 5F and 5G); NMDS analysis further revealed that I.F, I.S, and I.T samples were also separated from each other. Samples of Phase I.F had a higher relative abundance of the genus *Pseudoxanthomonas* than did those of Phase I.T ($P = 0.085$) and II ($P = 0.072$), as determined by the Wilcoxon signed-rank test (Fig. 5H). Samples of Phase I.F had a higher relative abundance of the genus *Thermobifida* than did those of I.T ($P = 0.088$) and II ($P = 0.065$). Samples of Phase II had a higher relative abundance of the genus *Thermobispora* than did those of I.F ($P < 0.05$) and I.S ($P = 0.073$).

LEfSe analysis identified the different microflora strains present within all samples (for LDA>3.5, Fig. 6A). The species of *Pseudoxanthomonas taiwanensis*, genera of *Bacillus* and *Sphingobacterium*, and families of *Paenibacillaceae* and *Beijerinckiaceae* were enriched in samples of Phase I.F. The family of *Rhizobiaceae* and genus of *Chelativorans* were enriched in Phase I.S samples. The species of *Pseudomonas fragi* and *Chryseobacterium hominis* were enriched in Phase I.T samples. The class of Acidimicrobiia was enriched in Phase II samples. A ternary plot was used to investigate the different bacterial strains of Phase I samples at the species level (Fig. 6B), which showed that *Pseudoxanthomonas taiwanensis*, *Ammoniibacillus agariperforans,* and *Ureibacillus thermosphaericus* dominated in I.F in comparison with that in I.T and I.S; *Pseudomonas fragi*, *Clostridium* sp. TG60.81, and *Thermobispora bispora* dominated in I.T compared to the other two Phase I groups. Comparing the dominant bacterial strains, we found that the relative abundance of *Thermobispora* and *Thermopolyspora* was higher in Phase II than in Phase I.

Moreover, the $t$-test was used to identify the different bacterial strains at the genus level between Phase I and II, which revealed the relative abundance of *Thermobispora*,

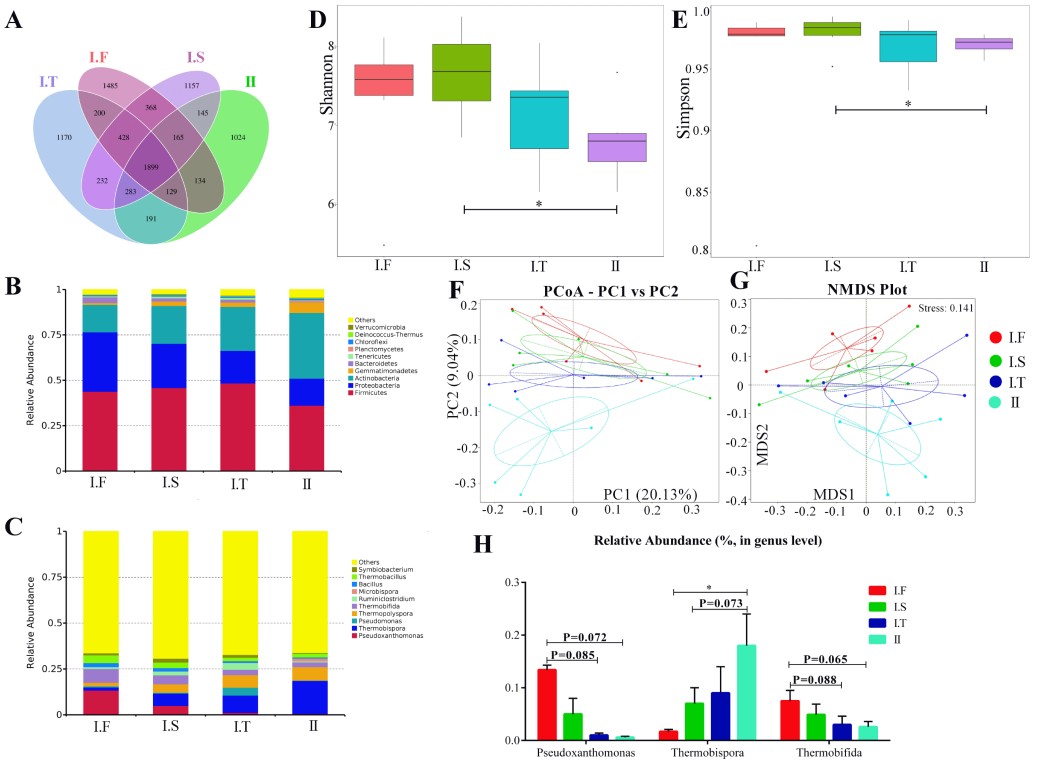

**Figure 5  Summary of the microflora community of compost samples in different phases.** (A) OTUs Venn diagram; (B and C) the microflora community of compost at the phylum and genus level; (D and E) the $\alpha$-diversity parameter (Shannon and Simpson indices); (F and G) PCoA and NMDS plot analysis of the microflora community of compost samples; (H) the comparison of the relative abundance of bacterial strains at the genus level.

*Thermopolyspora,* and *Vulgatibacter* to be significantly higher in Phase II than in Phase I.F (Fig. 6C). The samples of I.S had a significantly higher ($P$ <0.05) relative abundance of *Paenibacillus*, *Tepidimicrobium*, *Chelatococcus*, *Desulfotomaculum,* and *Flavobacterium* than did those of Phase II (Fig. 6D). The samples of I.T had significantly higher ($P$ <0.05) relative abundance of *Sphingobacterium* and *Defluviitalea* than did those of Phase II (Fig. 6D).

## DISCUSSION

The production of edible fungi has emerged as a potential biological use of agricultural AS, supported by low energy consumption and environmentally impact. Given the large variety of agricultural waste likely to be generated across China, the focus is mainly on the most prominent residues, such as rice straw, wheat straw, rye straw, and banana plant waste (*Zhang et al., 2019*). Typically, AS are treated as lignocellulosic agricultural byproducts, which comprise stalks, stems, and cobs. The use of agricultural straws in the edible fungi industry does not only take advantage of the waste of natural resources and reduce the environmental pollution derived from burning abandoned straws, but also has economic benefits (*Lu et al., 2018*). As a kind of white rot fungi, *A. bisporus* can degrade the majority of AS; in China, RS and WS have been identified as an important raw source for the

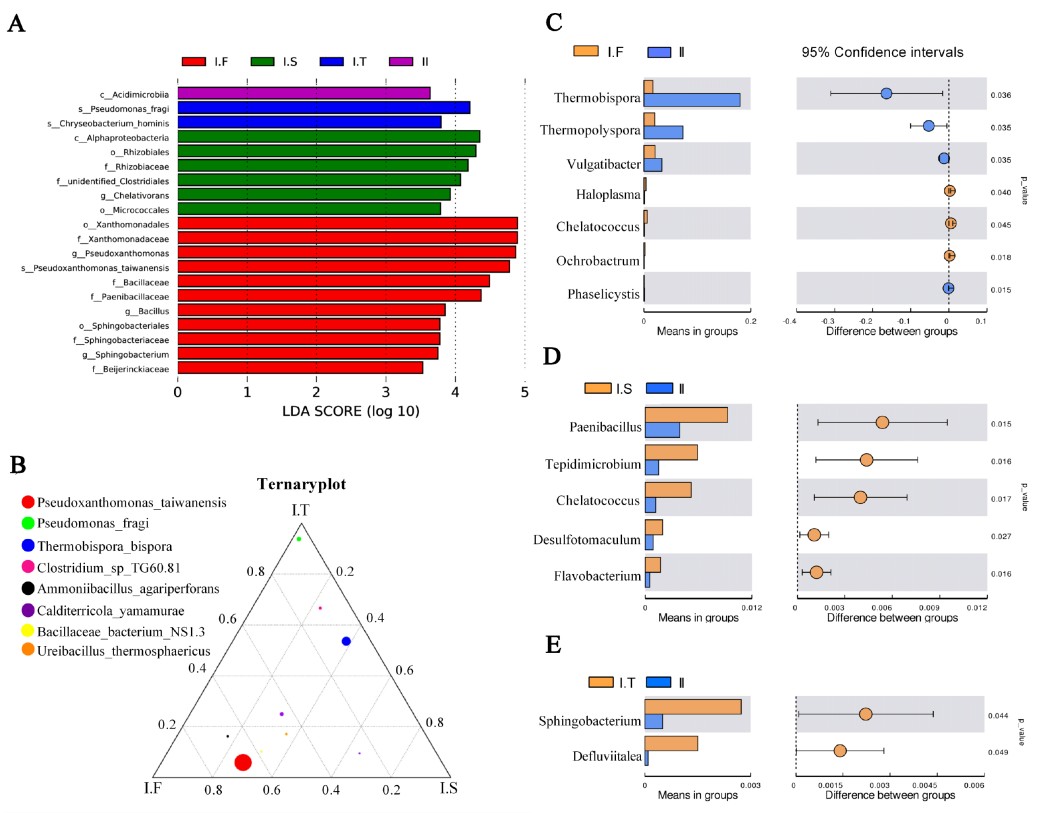

**Figure 6** **Analysis of microflora community structure differences in the two-stage composts of *A. bisporus*.** (A) LEfSe analysis for the four treatments; (B) a ternary plot about the I.F, I.S, and I.T treatments; (C, D, E) the t-test concerning I.F vs. II, I.S vs. II, and I.T vs. II, respectively.

substances of *A. bisporus* compost (*Wang et al., 2016*). In the current study that ranged from 2015 to 2017, the TY, T3, and EB parameters showed that the substances of RS, WS, and CS exhibited the highest mushroom yield, while C and B had poor yields under the same compost condition.

A large number of bacteria and fungi play a key role in the production of *A. bisporus* compost. They fermented AS into high quality substrate for *A. bisporus* by the thermophilic composting process. Those microorganisms decompose the compounds, release ammonia and then digest cellulose and hemicellulose into compost microbial biomass serves as the nutrition source of the *Agaricus* mycelium (*Kertesz & Thai, 2018*). *Sparling, Fermor & Wood (1982)* reported that wheat straw represents an important carbon source for *A. bisporus* cultivation, and that microorganisms play a key role in the degradation process. *Shen & Chen (2009)* reported that straw, containing abundant carbon, nitrogen, phosphorus (P), potassium (K), and other nutrients, is a valuable organic fertilizer and renewable resource. A study conducted by *Wang et al. (2016)* found that the C/N ratio in RS compost decreased more rapidly than that in WS compost. It has also been reported that the soil C/N and total N concentration are significantly increased during corn straw application (*Lu et al., 2015*). Here, no significant differences in the C/N ratio and N amount were observed; RS had a

higher N ratio than did CC in the Phase I.F, whereas CS and CC had a higher N ratio than did B in Phase II. The correlation analysis indicated that the N concentration in mature compost seemed to be related to the mushroom yield ($r = 0.78$, $P < 0.05$).

Microbial degradation of various agricultural residues leads to changes in physical and nutritional qualities, which can, for example, lead to changes in white rot fungi texture, degradation of lignin, and conversion of complex polysaccharides into simple sugars (*Sharma & Arora, 2015*). Here, the data revealed no significant differences in the crude fiber of substances between the initial formulation to I.T, and that in Phase II, CS had a significantly higher crude fiber content than did WS. Moreover, we determined the percentage of cellulose, hemicellulose, and ADL at the end of the study, revealing that after Phase II, CS, WS, RS, and CC after Phase II all had significantly less cellulose than did the other materials, and all compost samples had a significantly lower percentage of hemicellulose. Based on the above data, we suspected that the substance type of the compost influenced the digestive efficiency of lignocellulose, which may be related to the chemical composition, cellulose structure, antinutrients, and microorganism strains of the compost. Based on our data pertaining to the 17-day aerobic short-time fermentation, high N amount and crude fiber concentration, and low dry matter loss ratios are necessary for high *A. bisporus* production. For the present composting method, WS, RS, CC, and CS are probably better suited for *A. bisporus* production, while over-fermentation, or the low N content of C and B will lead to poor production of *A. bisporus*.

*Sheng et al. (2018)* showed that the structure of pretreated RS was severely destroyed, and more cracks and large holes were observed with the extension of fermentation time through scanning electron microscopy(SEM). *Shahryari et al. (2018)* found that high intensity fungal growth led to larger accessible surface area and structural destruction in the wheat straw observed by SEM. Similarly, in this study, TEM analysis showed that the cellular structure was destroyed by the degradation of *A. bisporus*. Interestingly, the destruction level of C and CC straw cellulose were higher than those in B and RS. Apparently, treatments with microflora not only improve the degraded biomass quality, but also simultaneously aid in the production of various commercial enzymes, which was connected to edible fungal production (*Sharma & Arora, 2015*). We speculated that the over-composting of CC and C caused the consumption of a large amount of nutrients and led to the relatively poor mushroom yield. Our previous studies found that the size of corn cob and corn straw affected the composting process and mushroom production under the same conditions (*Yu et al., 2018*). Hence, further investigations are required to elucidate if a more serious structural destruction of AS by the edible fungal pretreatment could lead to higher production.

It has been confirmed that *A. bisporus* is a classical example of sustainable food production, which is cultured on compost made from agricultural waste products. The edible fungal compost production is a process involving the bioconversion of raw materials into a composted mixture, of which the temperature increases to 80 °C owing to thermophilic microorganisms (*Zhang et al., 2019*). The activity of microorganisms alters the natural decomposition of lignocellulose in the compost of *A. bisporus* (*Jurak, Kabel & Gruppen, 2014*). The compost preparation (Phase I and II) of a high yielding

substrate is probably the most critical phase of mushroom production (*Sharma, 1991*). Composting is an organic materials decomposition process, in whichit has been confirmed that a range of successional taxa microbial convert wheat straw into compost in the thermophilic composting process, which is performed by a microbial organisms consisting of the thermophilic proteobacteria and actinobacteria, and thermophilic fungus (*Kertesz & Thai, 2018*). In the current study, the structures of microflora communities were found to change over the course of the fermentation time, based upon the outcome of the OTUs and dominant phyla and genera observed. It is worth mentioning that the Shannon and Simpson parameters indices showed that samples in Phase I.S possessed the highest bacterial community diversity. Moreover, considering the differences between microflora in composts, both the PCoA and NMDS analysis indicated that the species composition of Phase II samples were different from that of the Phase I samples.

*Xiao et al. (2018)* used high-throughput sequencing to reveal that Proteobacteria was the dominant phylum in thermophilic anaerobic fermentative stages. A previous study reported that Firmicutes, Proteobacteria, Actinobacteria, and Bacteroidetes were the most dominant phyla (*Vieira & Pecchi, 2018*). Similarly, in this study, the predominant phyla were observed to be Firmicutes, Proteobacteria, and Actinobacteria. The present study also indicated that the relative abundance of the genus *Thermobispora* increased along with fermentation time, while the genera of *Pseudoxanthomonas* and *Thermobifida* decreased. Our previous study (*Cao et al., 2018*) showed that *Thermobispora*, *Thermopolyspora*, *Ruminiclostridium*, *Thermobacillus*, and *Bacillus* were the predominant genera bacteria in Phase I.T and II in wheat straw consistent with the present study. Studies by both *Székely et al. (2009)* and *Zhang et al. (2019)* supported our data as they found that the genera *Thermopolyspora*, *Pseudoxanthomonas*, and *Thermobifida* were more abundance in mature compost. In addition, it has been reported that the genus *Thermopolyspora*, comprising a cellulolytic actinomycetes species, is highly enriched in mature mushroom compost based on wheat straw (*Zhang et al., 2014*). Our previous study also reported that the genera *Thermobispora* and *Thermopolyspora* were predominant in phases I.T and II of wheat straw substrate, by LEfSe and ternary plots (*Cao et al., 2018*).

## CONCLUSION

The results of this study demonstrated that six different AS exhibited differences in the progression of degradation under the same compost condition and the WS, RS, and CS produced a significantly higher mushroom yield by 16 day of composting. The genera *Thermobispora*, *Thermopolyspora*, and *Vulgatibacter* may play an important role in the different AS degradation. Our findings could be a clue to adjust formulations and compost methods to obtain high-yield mushroom compost cbased on different AS in the future. China is a large agricultural country, so the development of different AS from different areas as substrates for *A. bisporus* cultivation has the potential to reduce the cost of transporting raw materials and increase farming profits.

## ACKNOWLEDGEMENTS

Sincere thanks to Yongjiang Dai (the farmer of experimental base, Horticulture Institute, Zhejiang Academy of Agricultural Sciences) for help in this work.

### Funding

This work was supported by the China Agriculture Research System (CARS -20) and Substrate Utilization of Crop Straw (201503137). The funders had no role in study design, data collection and analysis, decision to publish, or preparation of the manuscript.

### Grant Disclosures

The following grant information was disclosed by the authors:
China Agriculture Research System (CARS -20).
Substrate Utilization of Crop Straw:  201503137.

### Competing Interests

The authors declare there are no competing interests.

### Author Contributions

- Tingting Song conceived and designed the experiments, analyzed the data, prepared figures and/or tables, and approved the final draft.
- Yingyue Shen analyzed the data, prepared figures and/or tables, and approved the final draft.
- Qunli Jin, Weilin Feng and Lijun Fan performed the experiments, authored or reviewed drafts of the paper, and approved the final draft.
- Guangtian Cao analyzed the data, prepared figures and/or tables, and approved the final draft.
- Weiming Cai conceived and designed the experiments, authored or reviewed drafts of the paper, and approved the final draft.

### DNA Deposition

The following information was supplied regarding the deposition of DNA sequences:

The 16s RNA sequences of compost for Agaricus bisporus production are available at GenBank: PRJNA664798.

BioSample: SAMN16233641, SAMN16233642, SAMN16233643, SAMN16233644, SAMN16233645, SAMN16233646, SAMN16233647, SAMN16233648, SAMN16233649, SAMN16233650, SAMN16233651, SAMN16233652, SAMN16233653, SAMN16233654, SAMN16233655, SAMN16233656, SAMN16233657, SAMN16233658, SAMN16233659, SAMN16233660, SAMN16233661, SAMN16233662, SAMN16233663, SAMN16233664.

The 16s RNA sequences of compost for Agaricus bisporus production are also available at Figshare:

1) SONG, TINGTING (2020): II.CS.fastq. figshare. Dataset. Available at https://doi.org/10.6084/m9.figshare.12764129.v1

2) SONG, TINGTING (2020): II.WS.fastq. figshare. Dataset. Available at https://doi.org/10.6084/m9.figshare.12764126.v1

3) SONG, TINGTING (2020): II.RS.fastq. figshare. Dataset. Available at https://doi.org/10.6084/m9.figshare.12764123.v1

4) SONG, TINGTING (2020): II.CC.fastq. figshare. Dataset. Available at https://doi.org/10.6084/m9.figshare.12764120.v1

5) SONG, TINGTING (2020): II.C.fastq. figshare. Dataset. Available at https://doi.org/10.6084/m9.figshare.12764117.v1

6) SONG, TINGTING (2020): II.B.fastq. figshare. Dataset. Available at https://doi.org/10.6084/m9.figshare.12764105.v1

7) SONG, TINGTING (2020): I.T.WS.fastq. figshare. Dataset. Available at https://doi.org/10.6084/m9.figshare.12764141.v1

8) SONG, TINGTING (2020): I.T.RS.fastq. figshare. Dataset. Available at https://doi.org/10.6084/m9.figshare.12764081.v1

9) SONG, TINGTING (2020): I.T.CS.fastq. figshare. Dataset. Available at https://doi.org/10.6084/m9.figshare.12764078.v1

10.) SONG, TINGTING (2020): I.T.CC.fastq. figshare. Dataset. Available at https://doi.org/10.6084/m9.figshare.12764075.v1

11) SONG, TINGTING (2020): I.T.B.fastq. figshare. Dataset. Available at https://doi.org/10.6084/m9.figshare.12764069.v1

12) SONG, TINGTING (2020): I.T.C.fastq. figshare. Dataset. Available at https://doi.org/10.6084/m9.figshare.12764072.v1

13) SONG, TINGTING (2020): I.S.WS.fastq. figshare. Dataset. Available at https://doi.org/10.6084/m9.figshare.12764057.v1

14) SONG, TINGTING (2020): I.S.RS.fastq. figshare. Dataset. Available at https://doi.org/10.6084/m9.figshare.12764054.v1

15) SONG, TINGTING (2020): I.S.CS.fastq. figshare. Dataset. Available at https://doi.org/10.6084/m9.figshare.12764048.v1

16) SONG, TINGTING (2020): I.S.CC. figshare. Dataset. Available at https://doi.org/10.6084/m9.figshare.12764042.v1

17) SONG, TINGTING (2020): I.S.C.fastq. figshare. Dataset. Available at https://doi.org/10.6084/m9.figshare.12764036.v1

18) SONG, TINGTING (2020): I.S.B.fastq. figshare. Dataset. Available at https://doi.org/10.6084/m9.figshare.12764030.v1

19) SONG, TINGTING (2020): I.S.B.fastq. figshare. Dataset. Available at https://doi.org/10.6084/m9.figshare.12764030.v1

20) SONG, TINGTING (2020): I.F.WS.fastq. figshare. Dataset. Available at https://doi.org/10.6084/m9.figshare.12764018.v1

21) SONG, TINGTING (2020): I.F.RS.fastq. figshare. Dataset. Available at https://doi.org/10.6084/m9.figshare.12764012.v1

22) SONG, TINGTING (2020): I.F.C.fastq. figshare. Dataset. Available at https://doi.org/10.6084/m9.figshare.12763997.v1

## Data Availability

There raw data is available in the Supplementary Files and at GenBank: PRJNA664798.

## Supplemental Information

Supplemental information for this article can be found online at http://dx.doi.org/10.7717/peerj.10452#supplemental-information.

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
