# Peer review of "Bacterial community diversity, lignocellulose components, and histological changes in composting using agricultural straws for Agaricus bisporus production"

_PeerJ, doi:10.7717/peerj.10452_

## Round 0.1 · original submission · Minor Revisions

Dear Dr. Cai,

Your manuscript was evaluated by two independent reviewers. Both reviewers have suggested changes to improve the manuscript. Please revise it based on these comments.

Sincerely,

Gunjan Arora, PhD

Reviewer 1 ·

Basic reporting

Song et al report a systematic and detailed study of degradation/composting of agricultural residues in term of lignocellulose composition, bacterial community diversity and morphological changers as a function of time. Study is very interesting and useful in understanding the bioprocessing of agricultural straws and usage of composting products in cultivating other edible/agricultural products including mushrooms.

The study is clear and unambiguous. Manuscript is clearly written.

Experimental design

Rigorous investigation performed to a high technical & ethical standard.

Methods described with sufficient detail and information to replicate.

Validity of the findings

All underlying data have been provided

Data is robust, statistically sound, and controlled

Conclusions can be improved by adding more detail about the application of the current study.

Additional comments

In the introduction section, I suggest adding more detail about the time required for composting preparation and how the current study could help a biotechnology industry.

Specific points:

1. It is very unclear how the temperature variation affects microbial community composition. Authors mentioned the microbial composition and temperature as a separate entity. A figure showing the “temperature vs biological community enrichment can be very helpful in understanding the importance of temperature in enriching a biological community diversity.
2. Figure 2. It is hard to follow so many parameters. Define “CS, WS, RS, CC, C and B” in the figure legend. Also define Y-axes N and C/N ration in the figure legend.
3. Table 1. Describe CS, WS, RS, CC, C and B.
4. Conclusions can be improved by adding more detail about the application of the current study.

Reviewer 2 ·

Basic reporting

A lot of grammar, spelling, and punctuation usage can be improved for example in line 35, 37, and 47-49. I would suggest getting the manuscript edited by a native English speaker.

Line 142: I was unable to find the reference cited here: Cao et al. 2019. Please make sure all the cited references are included in the References section.

Fig. 3 and Table 1: The usage of small alphabets statistical differences is hard to understand. Please use conventional asterisks or provide p values for each significantly different pair.

In the interest of readers from diverse background, I suggest expanding some of the abbreviations like C, N, P, K, RS, WS, CS etc. Use them in their full forms throughout the manuscript. It would also prevent mistake like “RC” instead of “RS” in line 289.

Experimental design

Lines 125-126: There seems to be a mistake in defining biological efficiency. It cannot be the ratio of T3 to TY. It should be the ratio of fresh mushroom weight to the weight of dry substrate.

Line 125: Elaborate if T3 is the weight of 1-3 flushes combined or is it average of all the three flushes. It may be helpful to include values for each of the three flushes in Table 1.

Validity of the findings

Line 266: Please elaborate what does this sentence signify “The N concentration in mature compost seems related to the mushroom yield”.

Additional comments

The manuscript by Song et al. describes the properties and characteristics of 6 different agricultural straws used for the production of mushroom. The authors compared different AS for their physico-chemical properties and analyzed changes throughout the fermentation process. They also sequenced and compared the microflora among these different AS. Based on their data, they recommend the AS from WS, RS, and CS to be most suited for mushroom production.
The results are exciting, however I have some minor queries.

Fig. 2A: It would be interesting to see a line graph of changes in C/N ratio among different stages of fermentation.

Line 33: The sentence is incomplete without mentioning how much AS are produced worldwide annually.

Line 44: please change “American” to “America”.

Table 1 and S3: Define “EB”. Is it same as biological efficiency (BE)?

Line 23: Please change “Phage” to “Phase”.

---

## Round 0.2 · Minor Revisions

Dear Dr. Cai,

The reviewer has some comments on the manuscript. Please revise the manuscript.

Reviewer 2 ·

Basic reporting

I have no further comments.

Experimental design

No comment

Validity of the findings

No comment

Additional comments

Most of the comments have been answered by the authors. I am however not satisfied with the line graph added to Fig. 2A. The line graph is for which of the 6 agricultural straws? Please elaborate or add line graphs for all the 6 AS.

---

## Round 0.3 · accepted · Accept

I am happy to accept the article based on the authors' response.